# Wafer-Scale Fabrication of Ultra-High Aspect Ratio, Microscale Silicon Structures with Smooth Sidewalls Using Metal Assisted Chemical Etching

**DOI:** 10.3390/mi14010179

**Published:** 2023-01-10

**Authors:** Xiaomeng Zhang, Chuhao Yao, Jiebin Niu, Hailiang Li, Changqing Xie

**Affiliations:** 1Key Laboratory of Microelectronic Devices & Integrated Technology, Institute of Microelectronics of Chinese Academy of Sciences, Beijing 100029, China; 2University of Chinese Academy of Sciences, Beijing 100049, China

**Keywords:** ultra-high aspect ratio, microscale silicon structures, MacEtch, sidewall roughness

## Abstract

Silicon structures with ultra-high aspect ratios have great potential applications in the fields of optoelectronics and biomedicine. However, the slope and increased roughness of the sidewalls inevitably introduced during the use of conventional etching processes (e.g., Bosch and DRIE) remain an obstacle to their application. In this paper, 4-inch wafer-scale, ultra-high aspect ratio (>140:1) microscale silicon structures with smooth sidewalls are successfully prepared using metal-assisted chemical etching (MacEtch). Here, we clarify the impact of the size from the metal catalytic structure on the sidewall roughness. By optimizing the etchant ratio to accelerate the etch rate of the metal-catalyzed structure and employing thermal oxidation, the sidewall roughness can be significantly reduced (average root mean square (RMS) from 42.3 nm to 15.8 nm). Simulations show that a maximum exciton production rate (G_max_) of 1.21 × 10^26^ and a maximum theoretical short-circuit current density (J_sc_) of 39.78 mA/cm^2^ can be obtained for the micropillar array with smooth sidewalls, which have potential applications in high-performance microscale photovoltaic devices.

## 1. Introduction

Over the past decades, micro/nano structured materials have found widespread applications in a variety of fields [1,2,3]. High aspect ratio (the ratio of Si thickness to lateral feature size) Si nanowires and micropillars have gathered an immense interest in micro/nano fabrication community due to their large surface area, which can be explored potentially in various fields, such as microelectronics [4], optoelectronics [5], photonics [6], photovoltaics (PVs) [7], and sensors [8,9]. However, in practical applications to photovoltaics, nanowire arrays always suffer from severe surface recombination due to excessive surface area and excessive defects generated during the fabrication process, which leads to reduced efficiency [10]. Recent works showed that the radial p−n junction can be more easily formed on the micropillars with relatively large diameters (large pitches) because they provide a good compromise between the effects of doping concentration, wire radius, and corresponding optical/electronic properties [11,12]. Thus, many researchers have endeavored to develop methods for the facile fabrication of Si micropillars and their applications [13,14,15].

Deep etching is an essential step in controllable Si microstructure fabrication [16]. It has become a powerful technique in the development and technological exploitation of advanced microelectromechanical systems (MEMS) [17]. Si deep etching can usually be categorized into two approaches: wet and dry etching (in liquid or gas phase, respectively). In the dry etching scheme, both Bosch (also called deep reactive ion etching) and cryogenic processes can be used for deep and narrow anisotropic etching with high etch mask selectivity [18]. However, achieving a straight sidewall with an aspect ratio larger than 50:1 still remains a considerable challenge. In addition, plasma-induced surface damage is unavoidable, especially since some surface defects inherent to the Bosch process (e.g., scalloped sidewalls), e.g., the scalloped depth of microscale width etched trenches, are typically up to 100 nm, which can severely degrade device performance [19]. On the other hand, the aspect ratio of etched trenches by anisotropic wet chemical etching methods (such as the KOH-based etching) is limited by the etching rate differences between different crystallographic orientations [20,21]. Although the accurate alignment of the lithographically defined mask to the crystalline plane is achieved, the aspect ratio is still limited at 80:1 [22].

As an alternative etching approach for fabricating Si microstructures, metal-assisted chemical etching (MacEtch) is operationally simple and cost effective [23]. Although MacEtch is anisotropic, methods have been developed to control the etch orientation, allowing the fabrication of vertically aligned silicon nanowires and micropillars on (100) and non-(100) substrates or certain tilted orientations on non-(100) substrates [24]. Additionally, the defects are less pronounced after etching, so the structure with high aspect ratio can be easily obtained by adjusting the process parameters (such as catalyst, substrate doping type, etchant formulation, etching additives, etc.) [24,25]. Various nanostructures, such as holes [26], pillars [27], grooves [28], and irregular three-dimensional structures [29,30], have been prepared by MacEtch. However, MacEtch also has its limitations when the size of the metal-catalyzed structures is on the micrometer scale. For example, scallop-like structures are introduced on the sidewalls, such as the result of Bosch etching, which affects the sidewall roughness [31]. Excessive sidewall roughness may lead to severe surface recombination due to the higher density of interface states, thus resulting in a poor sidewall passivation and ultimately degrading the performance of PV devices. Lots of methods, such as hydrogen annealing, IPA wet etching, and focused ion beam (FIB) milling have been carried out to reduce the sidewall roughness [32,33].

In this paper, ultra-high aspect ratio (>140:1), microscale Si structures with smooth sidewalls were successfully fabricated covering a four-inch wafer using MacEtch. We demonstrated the origin of sidewall roughness formation in fabricating microscale structures using MacEtch. The sidewall roughness can be significantly suppressed by accelerating the etching rate, which can be realized by optimizing the etchant ratio. To further reduce the sidewall roughness, the process of thermal oxidation and hydrofluoric acid rinsing was introduced immediately afterwards, and the average root mean square (RMS) of the sidewall roughness was reduced by 62.6% (from 42.3 nm to 15.8 nm). From FDTD simulation results, it can be seen that micropillars with smooth sidewalls have the ability to produce higher exciton generation rates and short-circuit current density, which are extremely important parameters for PV devices.

## 2. Materials and Methods

In a typical MacEtch approach, a metal layer (e.g., gold, Au) is patterned onto the substrate (e.g., Si) to locally increase the dissolution rate of the substrate material in an etchant solution, including a fluoride etchant such as hydrofluoric acid (HF) and an oxidizing agent, such as hydrogen peroxide (H_2_O_2_). H_2_O_2_ is reduced at the Au surface producing water and at the same time injecting holes through the Au catalyst into Si. At the Au–Si interface, the Si atoms underneath the Au catalyst are oxidized by the holes and dissolved in the HF solution as H_2_SiF_6_ [24]. Figure 1a shows the anisotropic etching fabrication process for high aspect ratio Si microstructures. This was started from a single crystal Si substrate coated with catalytic metal film and photoresist material. (i): 4-inch double-sides polished P-type <100>-oriented Czochralski (CZ) Si wafers with a resistivity of 1–20 Ω·cm and a thickness of ~500 μm were used as the starting substrates. The substrate was cut into 1 cm^2^ pieces, and ultrasonically cleaned in acetone and anhydrous ethanol for 10 min, followed by 5 min of washing in deionized (DI) water. A 3 nm Ti adhesion layer and a 20 nm Au layer were sputtered on a Si substrate by a magnetron sputtering system (ACS-4000, ULVAC Corporation, Japan), and the Ti/Au layer structure was used as an etching catalyst for the MacEtch process. Then, a single layer of negative photoresist (NR-1500) was spin-coated on the Ti/Au layer and baked on a hot plate at 150 °C for 2 min, and then it was masked with rows of 2 μm width lines spaced by 20 μm interconnected metal catalyst, which was used as a mesh to pattern the metal deposition. Second, we patterned the photoresist by using UV lithography in hard contact mode. (ii): Hard contact mode UV lithography and resist development were performed to define etching patterns onto the resist. At this step, the substrate was treated with mild oxygen plasma to remove the residual resist. After that, the photoresist pattern was transferred to the metal film using ion beam etching. (iii): The resist patterns were transferred to the Ti/Au layer by using ion beam etching for 60 s. The specific process parameters for ion beam etching were as follows: working pressure was −2 Pa, etching temperature was −15 °C, beam current was 90 mA, accelerating voltage was 160 V, and gas flow of argon was 17 sccm. Finally, after removing the photoresist, MacEtch was performed in the HF, H_2_O_2_, and deionized (DI) water to form anisotropic high aspect ratio micropillar arrays. (iv and v): The remaining resist was removed by acetone. Then, the samples were placed into the room temperature etchant to vertically etch off the Si beneath the Ti/Au interconnect network to form Si microstructures with high aspect ratios, with trench and micropillar widths of 2 μm and 20 μm, respectively. The etching time was 24 h, and the etchant was composed of HF, H_2_O_2_, and DI water in a molar ratio of 4.8:0.12:50. After the microstructure arrays were fabricated, the catalyst film of Au was removed by using a mixed aqueous solution of KI/I_2_/H_2_O (weight ratio = 4:1:10). The fabricated Si samples were characterized by optical microscopy (Olympus Model MX51) and scanning electron microscopy (Zeiss Supra VP55).

The key strategy of this approach is to use the interconnected metal catalyst mesh to form the high aspect ratio microstructures, which controls the etch directionality and uniformity. As shown in Figure 1b, due to the sufficient anisotropy and selectivity of ion beam etching, one can see the continuous and clear edges of the fabricated Ti/Au mesh. The interconnected Ti/Au catalyst mesh films are patterned with high density and high uniformity. High aspect ratio micropillars were successfully fabricated, and the sidewall profile was observed by the SEM (Figure 1c). The light and dark streaks were produced on the sidewalls of the micropillar corresponding to the undulation of height.

## 3. Results and Discussion

Figure 2 shows the mechanistic analysis of the formation process of sidewall roughness. Diffusion processes between metal, substrate, and the solution follow the mass transport (MT) model: diffusion of oxidized and dissolved Si and byproducts takes place in a thin permeable channel formed at the Si/metal interface, and the diffusion length is determined by the lateral size (distance between the openings) of the metal film. The etching solution has no way to penetrate across the metal thickness to dissolve the Si and eliminate the byproducts, except where the metal/Si interface ends at the edges of the patterned catalyst. In this case, the solution must pass different lengths that vary from 1 μm (half of the trench’s width) to several hundred nanometers. The formation process of sidewall roughness can be divided into four stages: (i) the metal layer maintains the same flatness as the substrate due to the presence of the adhesion layer. (ii) Small bending deformation of metal layer: the mechanical deformation of catalytic metal layer is induced by the uneven etch rate across the lateral dimension of the metal catalyst; there exists a faster etch rate at both ends of the catalyst layer compared to central region. (iii) The metal layer structure achieves the maximum bending degree. (iv) Due to the traction force of interconnected metal catalyst mesh, the metal layer gradually returns to flatness. Therefore, the metal structure morphology is in an alternating cycle of extension and contraction during the MacEtch process, which leads to greater sidewall roughness.

To compare the sidewall roughness and explore the relationship between roughness and the size from the metal catalytic structure, Si micropillar arrays with different trench widths were fabricated by using photolithography or electron beam lithography process combined with the MacEtch process. By adjusting the interconnected metal line width, micropillars with different trench widths ranging from 0.5 to 3 μm were fabricated, as shown in Figure 3a–d. 0.5 μm and 1 μm trenches (Figure 3a,b) were defined by electron beam lithography, and 2 μm and 3 μm trenches (Figure 3c,d) were defined by UV lithography. Sidewall surface morphologies were measured by atomic force microscopy (AFM), RMS roughness is shown in Figure 3e, and the roughness gradually increases as the trench width of the micropillar increases.

The line edge roughness (LER) of sidewall was characterized by a home-built software for image analysis of PSD and 3D Advance (Figure 3f). The intercepted SEM image was divided into 100-pixel points for each structure, and the change in gray level was used to represent the degree of sidewall roughness. Compared to the micropillars with 0.5 μm and 1 μm trench widths, the variation in sidewall gray level of the micropillars with 2 μm and 3 μm trench widths is significant, especially the largest variation with 3 μm trench width, which is consistent with the increase in RMS roughness (Figure 3e). The difference between these microstructures is mainly due to the MT [34]. The MT is not a critical bottleneck in the MacEtch of nanoscale Si structuring due to very short lateral distance (less than few microns) for the etchant/byproduct to travel [35]. The etching rate at both ends of the catalyst layer remains almost consistent with the central region, and the metal layer maintains the initial shape during the MacEtch process, as illustrated in Figure 3a,b. For the fabrication of microscale pattern features on Si, however, the MT should be the very important consideration. The difference in the etch rate between ends of the catalyst layer and central region is originated from the limited transport of etchant/byproducts to the central region of catalyst layer. Therefore, if the classical MacEtch used for Si microstructure fabrication without suitable design of process variables, it will lead to unsuccessful etching results due to deformation and bending of the catalyst layer (Figure 3c,d).

The effects of the concentrations of H_2_O_2_ and HF on the structure morphology were then investigated. Figure 4a–c show the SEM images of Si micropillar (20 μm width and 2 μm trench) etched with various H_2_O_2_ concentrations. The HF and DI water concentration were fixed at 4.8 and 50 M, and the H_2_O_2_ concentration was varied with 0.08, 0.10, and 0.12 M. The concentrations of H_2_O_2_ affect not only the etching rate, but also the morphologies of the etched structures. The etching rate monotonically increases as the concentrations of H_2_O_2_ increases. When the H_2_O_2_ concentration reaches 0.12 M, holes begin to occur on the sidewalls uniformly (Figure 4d,e), and the density of the holes will increase with the H_2_O_2_ concentration. As the role of oxidant is to inject h^+^, the increased concentration of H_2_O_2_ leads to an incremental rate of h^+^ injection [26]. However, for the injected h^+^, after the partial consumption in the oxidation of Si, the remaining part can diffuse out from the location of the metal catalyst. The Si surface away from the catalyst site was oxidized and dissolved by HF, leading to mesoporous Si structures [36]. When the concentration is further increased, structural destruction will occur due to the out-of-control hole diffusion during MacEtch. Figure 4g shows the LER image of micropillars sidewall with different H_2_O_2_ concentrations. The sidewall roughness decreases as the concentrations of H_2_O_2_ increases. When the H_2_O_2_ concentration was 0.08 M, the generated holes are mostly consumed at the edges of the interface between Si and Ti/Au, and only a few holes are accessible to central region, and this leads to the large roughness on the sidewalls. As the concentration increases, sufficient holes will be injected into the center of the interface, allowing the sidewall roughness gradually decreases. It is worth mentioning that the difference in etch rate between the center region and the edge will never be eliminated. Therefore, the morphology of the etched structure has a balance between lateral etching and sidewall roughness.

Microscale Si features prepared by MacEtch can be applied to solar energy conversion. Radial p–n junction solar cells based on the Si micropillars fabricated by optimized MacEtch showed enhanced efficiency compared to planar counterpart [7,37]. As shown in Figure 5a,b, the distribution of exciton generation rates and energy generation rates in the wavelength range of 400–1100 nm for Si micropillars with different roughness were analyzed by finite-difference time-domain (FDTD). FDTD simulations were carried out using the Lumerical FDTD software package. Two monitors were used to calculate the reflection spectra and the absorption spectra. The exciton generation rate distributions were simulated by the home-built program files. The Si substrate at the bottom was placed underneath the micropillar structures. Periodic boundary conditions were used to perform periodic array in the x- and y-directions. Perfectly matched layers (PML) were used along the z-direction. The Si material was modeled from the Palik material data provided by Lumerical. The simulations were performed in the wavelength range of 400–1100 nm using a plane–wave light source.

The distribution of exciton generation rate is an important factor in determining the photoelectric power conversion efficiency [38]. The distribution of exciton generation rate is similar according to Figure 5a, and the exciton generation rate gradually decreases as it extends from the central region. However, the maximum exciton generation rate (G_max_) is varied from different sidewall roughness of micropillars. As the roughness grows, the G_max_ presents a decreasing tendency. The micropillars with smooth sidewall were suitable for maintenance at a greater exciton generation rate distribution region (G_max_ = 1.21 × 10^26^), higher than that of micropillars with sidewall roughness of 50 nm (G_max_ = 1.03 × 10^26^) and 100 nm (G_max_ = 8.67 × 10^25^). The more excitons there are, the more electrons will be collected, leading to a higher photocurrent. Therefore, the photon absorption of micropillar arrays can be enhanced by controlling the sidewall roughness, thus increasing the light-trapping ability. Figure 5b shows the smooth sidewall micropillar arrays can effectively concentrate the light with greatest energy generation rates (G’_max_ = 28), further confirming the excellent light-trapping ability of smooth sidewall micropillars.

In addition, the FDTD simulations were also performed to simulate the absorption and reflection of different roughness micropillar arrays (Figure 6a,b). The smooth sidewall micropillars have the higher absorption (82.5%) and lower reflectivity (13.4%) at wavelengths of 400–1100 nm. In the short wavelength region <700 nm, c-Si has strong selective absorption [39], so these three types of Si micropillar arrays with different sidewall roughness have higher absorption. However, the surface recombination will be enhanced when the sidewall roughness grows, and the irregular sidewall pattern enhanced the diffuse reflection of the incident light, leading to lower anti-reflective ability and reduced absorption. On the other hand, as the wavelength increased, although the waveguide mode would become more pronounced, which is beneficial to improve the absorption of the Si structure [40], the Si has poor responsiveness to red and near-infrared light, thus increasing the optical loss and leading to the decrease in absorption. Theoretical J_sc_ was obtained by substituting the value of the simulated absorption of c-Si into Equation (1). By assuming that the internal quantum efficiency is 100%, irrespective of the wavelength, the term for the external quantum efficiency (EQE) was replaced by the absorption term [41,42].
(1)$$J_{sc}=\int_{400\;nm}^{1100\;nm}\frac{1.24}{\lambda }F\left(\lambda \right)\cdot EQE\left(\lambda \right)d\lambda$$
where λ is the wavelength of light and F(λ) is the solar spectral irradiance under AM 1.5 G according to American Society for Testing and Materials (ASTM) G-173. The J_sc_ values calculated based on the simulated absorption spectra are 35.26, 37.27, and 39.78 mA/cm^2^ for micropillars with 100 nm, 50 nm, and smooth sidewall roughness, respectively, which is attributed to increased numbers of surface defect states. Thus, controlling the sidewall roughness is an effective method of maximizing the light absorption by the c-Si micropillar arrays while minimizing the occurrence of surface recombination.

Figure 7 shows the high aspect ratio micropillars (20 μm width and 2 μm trench width) prepared under the optimum etchant ratio. As shown in Figure 7a,b, the AFM measurements were performed with a scan area of 10 × 10 µm^2^. The RMS roughness of the sidewall is 42.3 nm, and the undulating roughness ranges from −156.8 nm to 182.1 nm. Damage restoration was achieved by using thermal oxidation technique (annealing in oxygen at 1160 °C for 3 h). The O_2_ annealing was performed at a flow rate of 5000 sccm and N_2_ as a protective gas was performed at a flow rate of 3000 sccm. The oxygen element of the surface was further confirmed using energy-dispersive X-ray spectroscopy (EDS) to scan the elemental mapping of the sidewall surface after annealing, as shown in Figure 7c,d. The surface of sidewall is uneven, with some peak-shaped protrusions. In the initial stage of oxidation, the surface reaction rate of the Si-SiO_2_ interface plays a major role in the oxidation rate, the reaction rate of each point on the interface is basically the same, and the SiO_2_ will gradually extend to the interior of the Si pillars, but the shape remains unchanged. As the oxidation proceeds, the diffusion speed of the oxidant plays a major role because the oxidant molecules at various points in the external gas phase pass through the SiO_2_ layer to reach each point of the Si-SiO_2_ interface, and the diffusion distance required is different. The shorter the diffusion distance, the easier the diffusion. Therefore, the oxidation rate of each point on the Si interface is different, a point with a short diffusion distance may obtained a faster oxidation rate, as the oxidation process continues to consume a part of the Si, these peak-shaped protrusions will gradually be flattened. The phenomenon of roughness reduction by oxidation smoothing is best explained by the Gibbs-Thompson relation [43]:(2)μ(κ)=μ(∞)+γΩκ
where µ(κ) is the potential at a surface of curvature κ, µ(∞) is the potential at a flat surface, γ is the surface energy, and Ω is the atomic volume. The relation signifies that the chemical potential of a surface is dependent on the curvature of the surface. Essentially, the peaks of a rough surface (positive curvature) have a higher potential or reactivity than the troughs of the roughness (negative curvature). For the case of oxidizing a rough Si surface, the Gibbs-Thompson relation implies a differential reaction rate occurring on a rough surface that results in roughness amplitude reduction (i.e., smoothing) of the surface [44]. At this time, the SiO_2_ coating was removed by rinsing with HF, and the regained Si surface roughness will be decline, as shown in Figure 7e. The AFM measurements (Figure 7f) show a reduced RMS roughness of the sidewall of 15.8 nm, and the undulating roughness ranges from −56.8 nm to 69.1 nm.

An optical image of the fabricated high aspect ratio micropillar arrays on a 4-inch Si wafer is shown in Figure 8a. The surface image is completely black, confirming that the structure has a low surface reflectivity. Figure 8b shows the micropillar array (20 μm width and 2 μm trench width) with smooth sidewall, which was successfully etched. The depth was as high as 285 μm after the whole process was finished, achieving an aspect ratio up to 140:1.

## 4. Conclusions

In summary, 4-inch wafer-scale, ultra-high aspect ratio (>140:1) microscale silicon structures with smooth sidewalls are successfully prepared using MacEtch. The mechanism of the sidewall roughness formation process in the microstructure has been demonstrated in detail, and the impact of the size from the metal catalytic structure on the sidewall roughness has been highlighted. By optimizing the etchant ratio to accelerate the etch rate of the metal-catalyzed structure and employing thermal oxidation, the RMS of the sidewall roughness was reduced by 62.6% (from 42.3 nm to 15.8 nm). From FDTD simulation results, it can be seen that micropillars with smooth sidewalls have the ability to produce higher exciton generation rates (1.21 × 10^26^) and short-circuit current density (39.78 mA/cm^2^) in the broad wavelength region from 400 to 1100 nm, which have potential applications in high-performance microscale photovoltaic devices.

## Figures and Tables

**Figure 1 micromachines-14-00179-f001:**
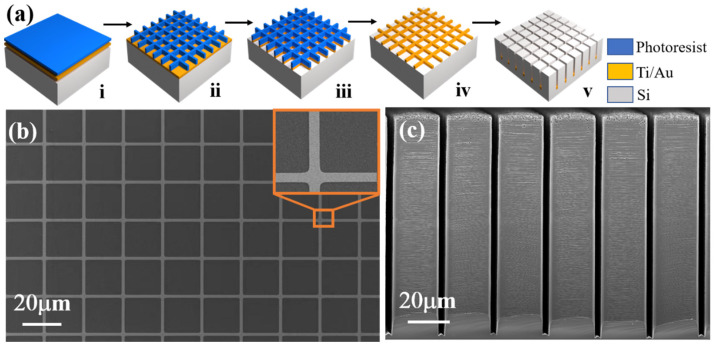
(**a**) Schematics of the fabrication process of ultra-high aspect ratio Si microstructures. (**b**) SEM images of the top micropillar arrays before MacEtch process. (**c**) Cross-sectional SEM images of high aspect ratio Si micropillars with sidewall roughness.

**Figure 2 micromachines-14-00179-f002:**
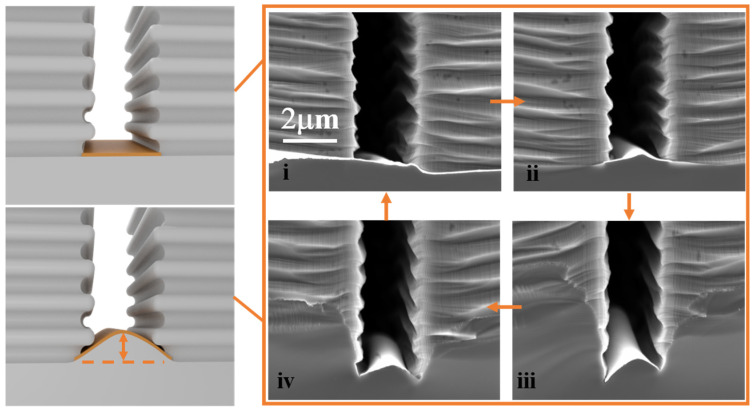
Mechanistic analysis of the formation process of sidewall roughness. (**i**) The metal layer maintains the same flatness. (**ii**) Small bending deformation of metal layer. (**iii**) The metal layer structure achieves the maximum bending degree. (**iv**) The metal layer gradually returns to flatness.

**Figure 3 micromachines-14-00179-f003:**
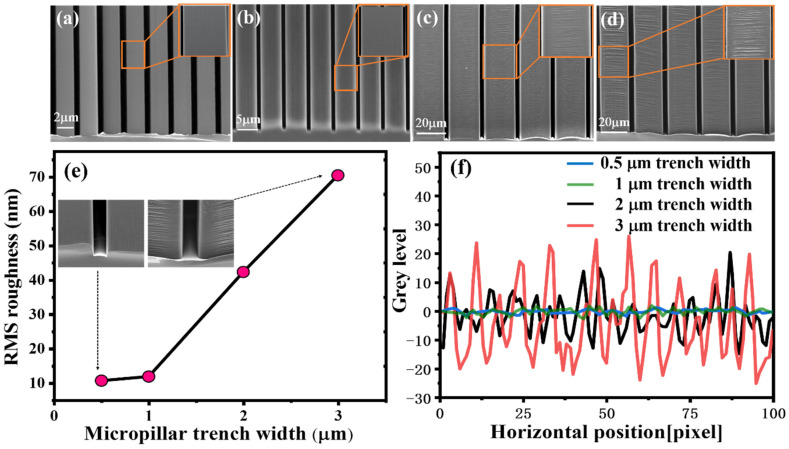
Cross-sectional SEM images of high aspect ratio micropillar trenches with (**a**) 0.5 μm trench width, (**b**) 1 μm trench width, (**c**) 2 μm trench width, and (**d**) 3 μm trench width. (**e**) The sidewall RMS roughness for micropillars with different trench widths. (**f**) The LERs image of micropillars sidewall with different trench widths.

**Figure 4 micromachines-14-00179-f004:**
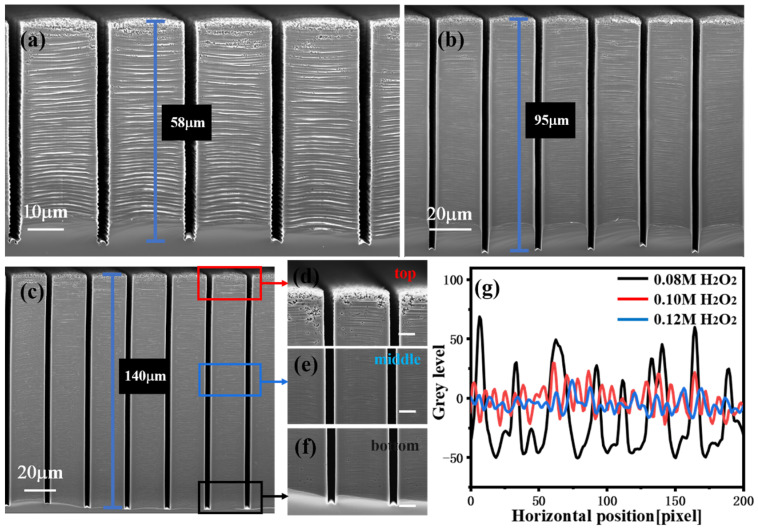
Cross-sectional SEM image of Si microstructures, etched under various etchant solutions composed of HF, H_2_O_2_, and DI water with molar ratio of (**a**) 4.8:0.08:50, (**b**) 4.8:0.10:50, and (**c**) 4.8:0.12:50. SEM images of the top (**d**), middle (**e**), and bottom (**f**) regions of (**c**). The etching time was 17 h, and the etching temperature was room temperature. (**g**) The LER image of micropillars sidewall with different H_2_O_2_ concentrations.

**Figure 5 micromachines-14-00179-f005:**
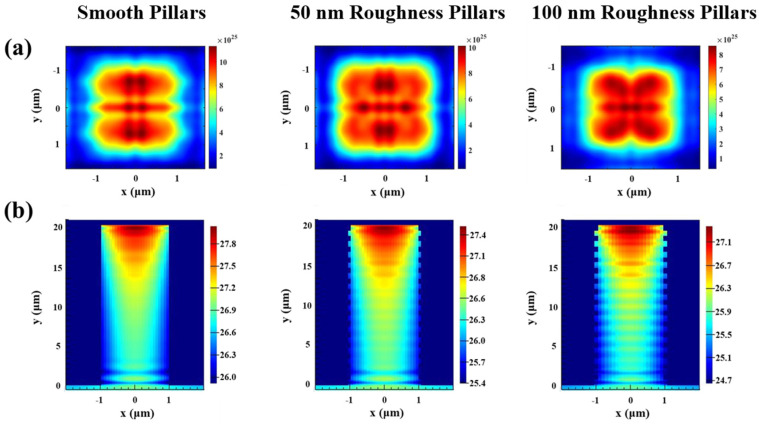
(**a**) Exciton generation rate distributions at the top of micropillars with different roughness at wavelengths of 400–1100 nm. (**b**) Energy generation rate distributions of micropillars with different roughness respectively at wavelengths of 400–1100 nm. The period of the micropillar is 4 µm, and the width is 2 µm.

**Figure 6 micromachines-14-00179-f006:**
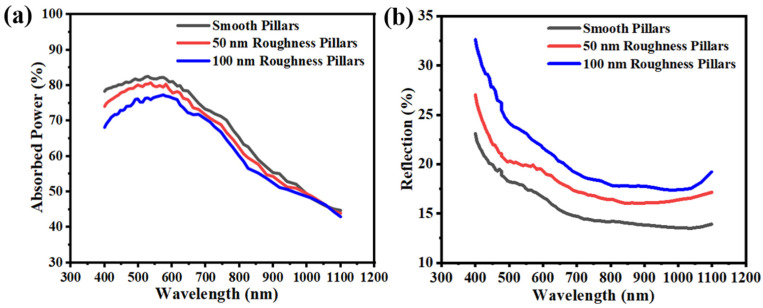
(**a**) The absorption spectrums and (**b**) the reflection spectrums of micropillar arrays with different roughness at wavelengths of 400–1100 nm.

**Figure 7 micromachines-14-00179-f007:**
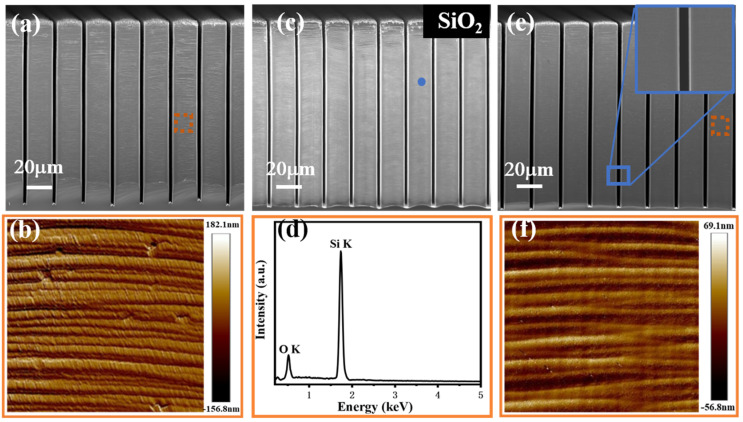
(**a**) The SEM image of micropillars before thermal oxidation. (**b**) AFM images of the orange box in panel (**a**). (**c**) The SEM image of micropillars after thermal oxidation. (**d**) EDS elemental mapping measurement of blue dots in (**c**). (**e**) The SEM image of smooth micropillars after removal of the oxide layer. (**f**) AFM images of the orange box in (**e**).

**Figure 8 micromachines-14-00179-f008:**
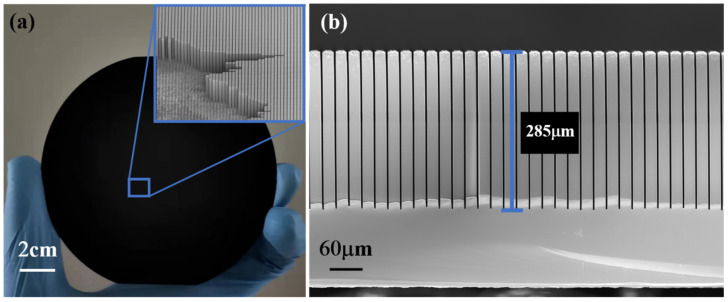
(**a**) An optical image of the fabricated high aspect ratio micropillar arrays on a 4-inch Si wafer, (**b**) The enlarged cross-sectional SEM images of ultra-high aspect ratio (>140:1) micropillars with smooth sidewalls.

## Data Availability

All data generated or analysed during this study are included in this article.

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
