# Peer review of "Wafer-Scale Fabrication of Ultra-High Aspect Ratio, Microscale Silicon Structures with Smooth Sidewalls Using Metal Assisted Chemical Etching"

_micromachines, 2023, doi:10.3390/mi14010179_

Round 1

Reviewer 1 Report

This study focuses on the fabrication of Si micro-structures with ultra-high aspect ratios using metal-assisted chemical etching and reducing the sidewall roughness by thermal oxidation and optimization of the etchant ratio. Improving the MacEtch quality of micro-structures is significant to micro/nanofabrication technology. I will recommend the publication of the paper after addressing the following questions.

1.    What about other processes available for machining high aspect ratio structures at present? The authors can describe the aspect ratio that can be achieved by other processes in detail.

2.    Please check whether the metal position in the schematic diagram in Figure 2 is correct.

3.    In Figure 3a-3d, when exploring the relationship between roughness and the size from the metal catalytic structure, trench widths ranging from 0.5 μm to 3 μm, but the spacing also increases with trench widths. Please clarify whether this would affect the result.

4.    The scale of each category in Figure 5 can be adjusted to the same, which is facilitated to subsequent analysis and easy understanding.

5.    Please describe the steps of MacEtch in detail. Was the voltage applied during the etching?

6.    Are the three pictures a-c in Figure 7 of the same size?

Reviewer 2 Report

The paper is focused on microfabrication by MACE as an alternative approach to standard RIE and DRIE fabrication. The paper is well written and it scientifically sound. There were already other works about the use of this approach but the study on the wall roughness it is very interesting and of interest.

My minor concerns:

1. I suggest the authors to introduce also sensing on the application field of silicon nanowires and pillars into the introduction as in the last years there were a lot of pubblications on that as (Nanomaterials 202111(7), 1767; Biosensors 202212(6), 368...)

2. line 54 pag 2 MACE is not limited by the crystalline orientation but for nanostructures is not as simple as the exposed number of atoms differs and tilted nanowires formation can happen as largely reported in literature. I suggest the author to discuss that point and to comment on the fact that I guess is not an issue for such big structures compared to nanowires.

3.  I get that point of the authors on the eq 1 pag 7 line 255. However, they are supposing a 100% internal quantum yield so this seems more a maximum Jsc with respect to a theoretical value. If this is the standard way to find it I ask the authors to introduce some ref about that. 

4. Even if the aspect ratio would not be the same the authors should report same value on the roughness tipically obtained by DRIE and RIE with a similar structure.

Only a comment for a future perspective. It would be interest to test the photovoltaic response of two identical same structures (in plane then you can play also on the higher aspect ratio that you can get by MACE) realized by MACE and DRIE to really show if MACE is worthed as an alternative.
